# Exploring the Experiences of Volunteer Assistance Dog Puppy Raisers from the Same Program at Two Australian University Campuses

**DOI:** 10.3390/ani13091482

**Published:** 2023-04-27

**Authors:** Sam Morwood, Dac Mai, Pauleen C. Bennett, Pree Benton, Tiffani J. Howell

**Affiliations:** 1Anthrozoology Research Group, Department of Psychology, Counselling and Therapy, School of Psychology and Public Health, La Trobe University, Bendigo, VIC 3552, Australia; 2Centre for Service and Therapy Dogs Australia, Melbourne, VIC 3162, Australia

**Keywords:** functional volunteerism framework, service dog, puppy socialisation, puppy walker, puppy foster carer

## Abstract

**Simple Summary:**

Puppy raisers (PRs) are central to the raising and training of assistance dogs (ADs). They are members of the public who volunteer to help look after an AD puppy for the first year of its life, in partnership with an AD program provider. Research has established that experienced PRs likely raise more successful puppies. What remains unexplored is the motivations and experiences of PRs as volunteers, who may return to raise more puppies if their program experience is generally satisfactory. In this study, we invited university staff and students from two campuses in Victoria, Australia, affiliated with the same university, to participate in a PR program endorsed by the university. We interviewed 16 participants throughout the program (three times for the Bundoora campus, and seven to ten times for the Bendigo campus) and analysed the data thematically to understand their experiences. We also considered our findings within a volunteer framework called the Functional Approach to Volunteerism. Our results supported past findings regarding the benefits PRs obtain from organisational and practical supports. We also identified four functional motivations for the PRs: altruism, egoism, interacting with others, and learning new skills. Future quantitative research is required to examine the relevance and influences of these motivations on PRs’ experiences and also on the training outcomes of their puppies.

**Abstract:**

Assistance dogs are beneficial; however, accessing one can be challenging due to the time, cost, and high failure rates associated with training. A major factor contributing to a high failure rate appears to be the competency of volunteer puppy raisers (PRs), with returning PRs typically more effective than first-time PRs. However, there remains a gap in the literature dedicated to examining PRs’ experiences and how they are affected by the raising programs. This study analysed two groups of PRs (*N* = 16) from the same university-based program in different Australian regions (i.e., one regional and one suburban). Semi-structured interviews were conducted and analysed both inductively and deductively. The inductive approach identified four major themes that helped to understand the PRs experiences throughout the program: expectations as a PR, organisational and environmental support and suitability, the intensity of PR workload, and practical support. The deductive analysis identified four functions of volunteerism relevant to PRs’ motivations: altruism (values), egoism (enhancement), interacting with others (social), and learning new skills (understanding). Overall, the results obtained from the two groups were consistent with past findings suggesting the benefits of organisational and practical support. These findings further develop our understanding of how to enhance puppy-raising experiences.

## 1. Introduction

Assistance animals help their handlers with a disability mitigate the impacts of that disability on their lives [1,2]. The term ‘assistance animals’ refers broadly to the role and training requirements of the animals instead of their species (e.g., equine or canine) [3]. Following specialised training, they can accompany their handler into public spaces prohibited to most animals [3]. ‘Assistance dog’ (AD) is an umbrella term for dogs specifically that are assistance animals and have been trained for one of several roles, including, but not limited to, guide, hearing, and alert dogs [3]. ADs assist individuals by increasing their physical independence, confidence, and improving their social relationships [4]. While ADs can be incredibly beneficial [5], and the number of disabilities they can help individuals with continues to increase [4,6], their widespread application is limited by the two or more years [7,8] and approximately AUD 40,000 (i.e., EUR 25,700; USD 27,000) [7,9,10] required to procure a fully trained AD.

To keep costs down for handlers, AD organisations typically recruit volunteer puppy raisers (PR) from the community to help raise each puppy for the first year of life. These PRs perform a vital community service, often at some personal cost. The inconvenience of caring for a puppy may result in PRs relinquishing the puppy during the program or not returning as a PR once the first puppy is returned to the organisation. Sustainable solutions for the AD industry include effectively outsourcing puppy caring and training responsibilities to PRs, which, in turn, requires understanding the experiences of these volunteers so as to improve recruitment and retention.

### 1.1. Puppy-Raising Programs

Raising puppies involves a long and demanding program that requires considerable sacrifice from volunteers. Volunteer PRs house puppies from around eight weeks of age to 12–16 months old [11,12], taking the puppies almost everywhere they go, including places off limits to non-ADs, such as supermarkets and work environments [5]. These activities are crucial to puppies’ behavioural development as they are aimed at increasing their socialisation opportunities and teaching them to enjoy being around humans and in novel environments, thus laying the foundations for developing the behaviour and confidence required of an AD [13,14,15]. However, the process is not always successful, with the high failure rate of puppies in these programs representing an industry-wide challenge. Data from the studies by Duffy and Serpell [16] and Tomkins et al. [10] of guide dogs in training found that 50–70% of the puppies were assessed as not suitable for advanced AD training programs, which all puppies must successfully pass before they are granted the access rights of ADs. This problem appears to be present in most AD training organisations [17,18,19]. 

A recent critical literature review [20] proposed that the most influential factors directly influencing AD puppies’ behavioural development are the PRs’ caring and training practices. Understanding the barriers to and enabling factors for engaging in appropriate training and caring practices may help AD organisations mitigate the negative impacts of raising programs on the PRs, whilst increasing the success rate of puppies in these programs. This is supported by some empirical evidence. Serpell and Duffy [14] conducted a longitudinal survey study examining (*N* = 978) AD puppies’ behaviour at 6 and 12 months old. At 12 months, several aspects of the puppy-raising environment correlated with the puppies’ behaviour, as reported by their PRs. Those who had been a PR before were more likely to raise a puppy that demonstrated positive behavioural outcomes and who was more likely to continue into advanced training. In contrast, the opposite was true for first-time PRs [14]. DeWitt [21] also found that those who had been PRs before were more likely to raise puppies who continued into advanced training, regardless of whether their first puppy was successful. The reason for this could be the increased experience returning PRs have over first-time [13,14,21,22]. Mai et al. [22] postulated that increased raising experience meant that PRs had had more opportunities to increase their knowledge base and practice their raising skills, resulting in increased competency. Therefore, it is reasonable to expect that success rates for puppies could be increased by retaining experienced PRs.

The high demands of being a PR, such as caring for and being with the puppies almost every day, can result in burnout [21,23]. Moreover, PRs must also teach the puppies the necessary skills to progress into advanced AD training [13]. To help PRs cope with these issues, the raising organisation commonly supplies training and informational support (e.g., regular training classes and trouble-shooting instructions). While existing efforts have aimed at easing the challenges of being a PR, e.g., [23], the results of enhancing benefits in order to retain experienced PRs are yet to be investigated. The cost associated with the puppy-raising program also includes the time and efforts of the puppy raisers, especially when volunteer PRs are not allowed to leave the puppies unattended for more than 3 h. Similar to other volunteer roles [24,25], AD organisations must highlight how the benefits outweigh the costs of the role for which the PRs are volunteering. Owing to this, the relevant literature on volunteerism may inform research and the AD industry regarding what motivates prospective volunteers to sign up in the first place, as well as to continue being a PR.

### 1.2. Potential Application of the Functional Approach to Volunteerism Framework to Understanding PRs’ Motivations

The Functional Approach to Volunteerism (FAV) framework [26] posits that when individuals’ motivations align with the benefits offered by a volunteering opportunity, they are more likely to engage in the first place and further participate in the future. The FAV describes six functional reasons that explain ways in which individuals can benefit from volunteering: values—expressing altruistic and humanitarian values; enhancement—serving positive aspects of the ego concerned with personal growth; social—enhancing or forming new social relationships; understanding—gaining and enhancing knowledge; career—help progressing their career; and protective—engaging in protective processes to protect their ego.

Of the six functional reasons, the understanding and social factors could help explain the experiences of PRs in recent research in the AD literature. Regarding understanding, Chur-Hansen et al. [23] found preliminary qualitative evidence of the consequences of a lack of organisational support on PRs’ program experiences. Chur-Hansen, Werner [23] interviewed nine PRs from the same program, once before receiving their puppy and three times throughout the 13-month study. Their principal findings were that the inconvenience of being a PR far outweighed the PRs’ perceived benefits due to insufficient organisational support. This resulted in PRs having to guess how they should raise, train, and care for the puppy, which they all struggled with as first-time PRs. This is supported by the findings from a quantitative survey study by DeWitt [21] with 499 PRs, which suggested that being a PR was substantially more challenging than initially expected. However, support for raising the puppy was rated as helpful. It was noteworthy that the most beneficial support did not come from the organisation but from household support (i.e., family members or partners). Those household members helped train, socialise, and care for the puppies [21]; similarly to Chur-Hansen, Werner [23], PRs who lived alone likely struggled as they were left to raise the puppy predominantly by themselves.

A notable aspect of DeWitt’s [21] findings was the socialisation aspect of being a PR, which was consistent with the social factor of the FAV framework. Specifically, it was reported that the AD organisation helped facilitate interactions amongst the PRs, helping them feel a part of the raising community and form friendships with other PRs. It significantly predicted their willingness to return as PRs, so DeWitt [21] argued that organisations wanting to retain PRs should strive to foster a sense of community within their raising groups. However, this would likely be a difficult task considering that PRs are usually drawn from the wider community and the levels of interaction between fellow members vary amongst regions, with individuals in regional areas more likely to socialise with and offer support than those in urban communities [27,28]. A raising program in which volunteers were located near each other, either in a regional/rural or urban/suburban area, would enable examination of the ‘social’ aspect of the FAV framework, in conjunction with the other five functional motivations. 

### 1.3. The Current Study

This study aimed to understand the experiences of PRs in the same raising program but located in two different parts of Victoria, Australia. One area was Bundoora, which is a suburb of Melbourne, a large city of 5 million people. The other area was Bendigo, a small city of 100,000 people about 150 km north of Melbourne. To achieve these aims, this study adopted an in-depth qualitative interview method to explore the impacts of the provision of support or the lack thereof during an AD puppy-raising program. Additionally, the specifics of the type of support that impacted the PRs are included in the analysis so future programs can begin to understand how to best support their PRs. Since the programs were implemented at two university campuses, a secondary aim of this study was to understand the suitability of campus environments for running AD-raising programs. The findings of this study will hopefully help future PR programs to improve their PRs’ overall program outcomes, with the long-term benefits for the AD industry including more satisfactory volunteer experiences and thus enticing them to raise further puppies.

## 2. Materials and Methods

The data in this study were collected as part of a larger project, i.e., [29]. The project received approval from the La Trobe University (LTU) Human Ethics Committee (HEC18-325 and HEC19-031) and the Animal Ethics Community, La Trobe University (AEC18043). 

### 2.1. Participants

We recruited sixteen LTU members (twelve women and four men), eight from the Bundoora campus (ethics approval number HEC19-031) and eight from the Bendigo campus (ethics approval number HEC19-031). Eight were undergraduate or postgraduate students, and eight were staff members holding various academic and non-academic positions. Participation in the program was voluntary, and participants were recruited via convenience sampling. Inclusion criteria required participants to be continuing LTU students or staff members for at least the next 12 months, at least 18 years of age, and able to communicate in English. After expressing interest in the study, the program’s AD training provider assessed the individual as having suitable personal characteristics and home and work environments to raise a puppy. There were no explicit exclusion criteria. Some participants had prior experience raising dogs and other pets; however, none had raised an AD puppy.

Although the recruitment of this study did not determine data saturation as a point of terminating data collection, our sample of 16 exceeded the recommended number of 12 for an exploratory qualitative study [30,31].

### 2.2. Materials

Participants were interviewed multiple times before, during, and following their involvement in the program. Semi-structured interview schedules were used with broad and non-leading questions, allowing conversations to unfold while staying on topic [32]. The questions focused on the PRs’ experiences and what they perceived as helpful or unhelpful aspects of the program. Before the participants received an AD puppy, questions in the initial interviews focused on their motivations for joining the program, their perceived suitability, and their expectations of the program, e.g., “Tell me why you would like to become a puppy raiser”. Subsequent interviews asked how the PRs’ experiences changed since their previous interview, e.g., “Could you tell me about your experience raising the puppy in the last month?”. The semi-structured interview schedules are available online in Mai et al. [33].

### 2.3. Procedures

The overall procedure of this study adopted an extended fieldwork (longitudinal) design to enhance theoretical validity [34]. A method of triangulating the data from different groups in the same program (i.e., Bundoora and Bendigo campuses) was employed to aid the internal validity of the study [34,35].

Because implementing a puppy-raising program as part of the university’s approved activity was not widely known, the program was piloted at the smaller campus, i.e., Bendigo, before being rolled out to the Bundoora campus. The two programs followed the same puppy-raising protocols of the AD provider. However, the research designs varied slightly to reflect the piloting and normal implementation nature of the overall project; there were also different research ethics approvals for the two cohorts. Specifically, during the piloting phase, we conducted more frequent interviews (seven to ten interviews for the Bendigo PRs), while the program at the Bundoora campus followed a more common research design with interviews conducted at three time points. 

In terms of recruitment, the program was advertised to university students and staff through official LTU media and communication channels from December 2018 to March 2019 for the Bendigo campus and July 2019 and October 2019 for the Bundoora campus. Since this was the first official puppy-raising program between a university and an AD program provider in Australia, the roll-out of the same puppy-raising program at two campuses was planned at different stages to account for the availability of puppies and staffing of the program provider.

Prospective participants received a participant information and consent form, and upon returning a signed copy, were asked to attend an initial interview before being admitted into the puppy-raising program by the program provider. Throughout the program, PRs attended several follow-up interviews over seven to fourteen months.

Once they received an AD puppy from the program provider, PRs were responsible for caring for and implementing training programs for those puppies with support from the AD organisation. This was in the form of material help, such as providing food, toys, collars, leads, and harnesses, as well as informational and training support for the PRs and their puppies. There were weekly, hour-long training sessions on each campus run by the AD organisation. Additionally, each PR was assigned one or more puppy socialisers, volunteers who would help train, socialise, and look after PRs’ puppies. For instance, each week, a socialiser may oversee their assigned puppy for periods ranging between 30 min to an entire day, thereby allowing the PRs a break from their raising duties.

As shown in Table 1, PRs in the Bendigo group attended between six to ten interviews, whilst Bundoora PRs attended three interviews. All interviews were conducted in person or via teleconference by the second author (D.M.). The interviews were audio-recorded and between 7 and 63 min long, with the Bundoora and Bendigo average times being 30 and 27 min, respectively. Four interviews from the Bendigo campus (i.e., HEC18-325) that were pseudo-randomly selected for analysis and included in the current study were also reported in Mai et al. [33].

The average number of days between Bundoora PRs receiving their puppy and attending their final interview was calculated to facilitate a comparison between campuses. The Bendigo interviews closest to this number were selected for analysis (refer to Table 1). For instance, Bendigo interviews conducted less than four weeks into the program (e.g., W1 or W2) were not used, as there were no Bundoora interviews conducted during this period. Four selected Bendigo interviews were omitted from the analysis due to missing audio files. Recordings of the 40 selected interviews were transcribed and analysed.

### 2.4. Data Analysis

Data analysis was conducted both inductively and deductively with the use of NVivo, a qualitative data analysis application [36].

#### 2.4.1. Inductive Thematic Analysis

Inductive thematic analysis [37] involves six phases to identify common themes across a dataset. Themes are meaningful and reoccurring patterns that can be used to address a study’s research question, such as understanding the experiences of PRs from the two LTU campuses in the current study.

Phases one to four involved familiarisation with the dataset through repeated readings, and a series of free coding attempts were undertaken, during which themes began to emerge. The first author performed the initial analysis, and then all authors discussed the analysis in-depth to identify areas of agreement and disagreement. This process of analysis and discussion was repeated until the research team reached full agreement on the codes, which were subsequently categorised into five major themes with eighteen sub-themes. The involvement of multiple researchers to understand and discuss a dataset was additional to Braun and Clarke’s [37] procedure. This form of investigator triangulation was utilised to reduce the risk of researcher bias, thus increasing the study’s descriptive validity [34,38].

Phases five to six involved defining themes and selecting quotes that illustrate the essence of each sub-theme. As Braun and Clarke [37] suggested, using direct quotes also helped ensure the interpretive validity of the findings, as the quotes gave voice to the participants [34]. However, commonly spoken words such as “umm” and “ohm” were omitted to aid readability.

#### 2.4.2. Deductive Qualitative Analysis

Deductive qualitative analysis refers to using a theoretical framework to analyse a dataset, providing a more detailed analysis of some aspects of the data [37]. Clary et al.’s [26] FAV framework was used in this study. Clary, Snyder [26] identified six core functions of volunteering that were used to inform deductive data analysis in the context of puppy-raising; this provided an in-depth understanding of how the PR’s motivational needs were met, or not, across the two campuses. After the bottom-up inductive data analysis following Braun and Clarke’s [37] procedure explained above, several observed codes were assessed for their compatibility with the components of the FAV. Therefore, they were coded as subthemes, with ‘motivations to participate’ being a main theme. The use of illustrative quotes was also utilised for the FAV subthemes.

## 3. Findings

The qualitative data analysis revealed five themes: expectations as a PR, organisational and environmental support and suitability, intensity of PR’s workload, practical support, and motivations to participate. Each theme has two to five sub-themes. Summaries of the themes and sub-themes are in Table 2, along with illustrative quotes from the participants.

### 3.1. Expectations as a PR

In the initial interviews, PRs from both campuses mentioned anticipating challenges in the program, namely that the puppies would make them emotionally and physically tired. However, they all expected the benefits of having a puppy to outweigh the costs of being a PR. All PRs expected the organisation to tell them what was expected of them (i.e., what they should be teaching and exposing the puppy to), provide them with training techniques (i.e., how they should be teaching the puppy), and supply informational support (e.g., trouble-shoot problems, such as puppies toileting inside or pulling on leads when walking). Additionally, most PRs expected support from their household members and work colleagues.

### 3.2. Organisational and Environmental Support and Suitability

In most interviews, PRs expressed how they had received, or not received, adequate support from the organisation. PRs mentioned more problems stemming from a lack of organisational support in their second interviews than in their third interviews. The expressed unmet expectations of organisational support might be due to higher demands for managing the puppies’ behaviours in the earlier stage (second interview) or it could be that the PRs had higher expectations of the organisational support which they might have re-calibrated as they gained more experience in this role (third interview). Most PRs often expressed disappointment with themselves and their puppies if the puppy misbehaved or their behaviour was not to the standard the PR wanted or expected. Training and everyday activities with the puppy were more enjoyable for the PR when the puppy behaved.

The university environment was considered a suitable raising location, as many PRs stated how helpful it was to have a populous training environment nearby. However, a few PRs from both campuses reported not enjoying the extra attention the puppy brought them. Only one PR from the Bundoora campus reported having a substantially negative experience with a colleague due to the puppy. When the puppy brushed against their leg, they immediately began complaining to senior management instead of discussing it with the PR. While LTU senior administrators had released a memo of support for this program, the incident with the colleague caused the PR substantial distress.

Regarding environmental suitability, the broader Bundoora and Bendigo communities predominantly welcomed the puppies. Misunderstanding AD laws were the cause of most problems; however, PRs could usually explain the puppies’ access rights and resolve the issues with minimal difficulty. Unfortunately, three PRs from the Bundoora and one from the Bendigo campus reported having problems with members of the public regarding their puppies’ access rights. For instance, one Bundoora PR reported being continually denied access by bus drivers despite explaining the AD puppies’ access rights.

### 3.3. Intensity of PRs Workload

Although PRs from both campuses reported issues with the puppies being a time-consuming task on top of already busy lives, the Bundoora PRs reported this more often than their Bendigo counterparts. Additionally, most Bundoora PRs reported being too busy to dedicate the necessary time for training and caring for their puppies, while only a few Bendigo PRs mentioned this. The reasons for this discrepancy are not clear from the available data. However, PRs from both groups felt they needed more support from the organisation. PRs from both campuses noted that learning to accept that their puppies would misbehave was a relief, as it reduced worry and lessened the burden that they put on themselves to unrealistically raise the perfect puppy.

### 3.4. Practical Support

Competent socialisers, a source of practical support, helped expose puppies to new environments and experiences regularly, allowing PRs from both campuses to take breaks from being PRs. In the current study, socialisers were recruited from the university community (i.e., staff and students), who wished to be involved but did not have a suitable living environment to raise the puppy or simply could not provide long-term care for the puppy. They attended training with the PRs and were coordinated by staff from the AD provider to take over socialisation activities for the puppies around the campus for a couple of hours per week while the PRs were working or studying on campus. Most PRs found the socialisers very helpful. However, PRs reported less competent socialisers as becoming a hindrance. In this regard, the Bundoora PRs reported more frustration with managing their socialisers than the Bendigo PRs.

PRs provided emotional support to each other by sharing their personal experiences and advice. These benefits extended to the training of the puppies, as they had more opportunities to interact and train together outside the weekly formal sessions with the raising organisation’s AD trainers. Furthermore, almost all PRs reported no issues with housemates and work colleagues. Instead, most PRs reported that individuals in their homes and workplaces often actively assisted in training, taking the puppy for walks and even caring for them when the PRs took some time off or visited places where they did not want to bring the puppies.

### 3.5. The Functional Approach to Volunteerism

#### 3.5.1. Values

In most interviews from both campuses, the PRs echoed similar sentiments regarding how they thought they were doing good by participating in the program. Some PRs from both groups shared how this helped keep them motivated.

#### 3.5.2. Enhancement

Most PRs from both campuses mentioned that being a PR made them feel good about themselves. However, these statements refer more to their altruistic intent and a sense of personal growth rather than acting as a protective process to reduce negative feelings resulting from guilt of being more fortunate than others. Most PRs from both groups shared how this helped maintain their motivation.

#### 3.5.3. Social

PRs developed new relationships with other PRs and could engage with more individuals in their personal community (i.e., household and friend groups) and broader community. PRs from the Bendigo campus reported considerably more social benefits from interacting with other PRs than the Bundoora PRs.

#### 3.5.4. Understanding

In most interviews, PRs from both campuses expressed that learning new skills while raising their puppies made their experiences more enjoyable. However, when PRs felt they were denied these opportunities through a lack of organisational support, their motivation was negatively impacted.

#### 3.5.5. Career

Only one PR mentioned their participation in a puppy-raising program as a way to advance their career, as it is an activity looked favourably on by others. Career advancement may be a motivational reason for joining the puppy-raising program in some cases.

#### 3.5.6. Protective

No PRs conveyed joining the program or receiving any benefits observed as acting as a protective process in a desire to protect their individual egos.

## 4. Discussion

This study analysed qualitative data to understand the experiences of puppy raisers (PR) in a university-based raising program, in partnership with an assistance dog (AD) training provider, at a suburban and a regional campus in Victoria, Australia.

### 4.1. Key Findings

There were several benefits and challenges that PRs expected to experience; however, all PRs in the current study believed that the organisation would help them minimise the difficulty of being a PR. To overcome the challenges of raising a puppy, PRs expected the organisation to provide adequate raising support and informational advice. This is consistent with past research, suggesting PRs anticipated similar costs and benefits for raising an AD puppy and relied on the organisation to assist them [23]. Furthermore, both Chur-Hansen’s [23] and DeWitt’s [21] findings demonstrated that PRs were motivated by largely the same altruistic and positive egoistic reasons as in our study.

Generally speaking, to increase PRs’ program satisfaction, the perceived benefits of organisational training and informational support need to outweigh the perceived costs of raising an AD puppy. However, the current findings revealed that for PRs to realise these benefits, the levels of organisational support must also exceed the PRs’ expectations. This added requirement of support was due to the first-time PRs’ tendency to underestimate the difficulty and complexity of raising an AD puppy. While this aligns with Chur-Hansen’s [23] findings, it is also concurrent with research highlighting the importance of organisations supporting volunteers in complex and demanding roles. For instance, Ross, Greenfield [39] found that AIDS volunteer workers dissatisfied with the lack of adequate organisational training support were more likely to experience distress and burnout and subsequently cease volunteering.

Our findings suggest that PRs wanted organisational support to increase their competency, which would reduce the difficulty of training and enable them to better manage their puppy’s behaviours. This supports DeWitt’s [21] finding that more competent individuals who had raised multiple puppies reaped more benefits of being a PR, as the challenges overshadowed their experience less. On the other hand, PRs throughout Chur-Hansen’s [23] program reported not receiving enough organisational support.

In addition to organisational training and informational support, practical support was reported as beneficial, as it reduced the PR’s workload and helped them enjoy their experience in the program. In this study, practical support from household members, work colleagues, and volunteer socialisers helped reduce the intensity of raising an AD puppy. For instance, household members would often look after the puppy, allowing the PRs to relax and partake in leisure activities independent of their puppies. As shown by Moreno-Jiménez and Villodres [40], individuals need to take breaks from continuous volunteering, as overworking themselves increases the likelihood of experiencing burnout, especially if they have paid jobs in addition to volunteering. They also found that those motivated by more altruistic than egoistic reasons were less likely to experience burnout. It is probable that this helped maintain PRs’ motivation in our study, as altruism was a major motivating factor for all PRs. However, the task remains challenging throughout the duration of the program, hence the requirement for ongoing support to avoid PRs experiencing burnout.

Household help greatly assisted PRs in DeWitt’s [21] study, whereas Chur-Hansen, Werner [23] found assistance from household members was present but predominantly not beneficial. Our study found that multiple different sources of help were beneficial for avoiding burnout, because multiple means of practical support (i.e., socialisers, household members, and work colleagues) meant some form of support was always available. For instance, several PRs’ household members assisted more when socialisers were unavailable due to being away for university holidays.

Individuals who shared their puppy-raising experience with fellow PRs benefitted from the emotional support and the enjoyment experienced from those interpersonal interactions. The emotional support PRs received from sharing their experiences with other PRs helped them deal with the challenges. Smith, Drennan [41] conducted a study using semi-structured interviews to examine carers (i.e., individuals who provide unpaid care and support for family and friends) for dementia patients and found similar results to our study. The findings suggested that when carers could share their negative experiences (e.g., when patients cried in front of them) they did not feel alone with their problems, helping them cope with the challenging situations and circumstances more effectively.

The benefits of having a shared understanding with others were more apparent during the current campus-based puppy-raising programs, especially when it came to the legal understanding of public access rights of the assistance animals and those in training. The PRs in the current study reported positive experiences when interacting with colleagues and the university community during socialisation and training of their puppies, even though they reported being denied access when conducting those activities in public places. Although public access for ADs in training varies by jurisdiction [42], in many parts of Australia, including the state of Victoria, public access for AD puppies is permitted [43]. Nonetheless, PRs commonly reported that general community misunderstanding about access rights of AD puppies in training was deterring and discouraging [44]. Overall, with the convenience of conducting puppy training and socialisation at the workplace (for staff) and in learning environments (for students), along with free access to short-term carers (i.e., volunteer puppy sitters or socialisers), university campuses appear to be a highly suitable environment in which AD provide should consider operating their puppy-raising programs. However, the puppy-raising program was fully supported by La Trobe University, which created a welcoming environment that may not be possible without full endorsement by university administrators.

#### Understanding PRs Motivation through the Functional Approach to Volunteerism

The experiences of PRs in this study were relevant to four of Clary et al.’s [26] FAV functions: values, enhancement, social, and understanding. Specifically, the current findings revealed that PRs were highly motivated to join the program for altruistic reasons, specifically to help someone in need of an AD and give back to the community, as reflected in Chur-Hansen, Werner [23] and DeWitt [21]. Moreover, egoistic reasons noted in our study, such as thinking that the puppy would bring positive character change to the PRs or that they would enjoy having the puppy in their lives, were also motivating factors in both Chur-Hansen, Werner [23] and DeWitt [21].

Regarding the ‘social’ function, our findings revealed that some PRs were able to achieve some expected benefits, such as increased cohesion between families and partners, as they worked together in raising the puppy. Additionally, interacting with the public was enjoyable or tolerable for most PRs, but some from both the Bundoora and the Bendigo groups reportedly found that people stopping them so they could interact with their puppy became irritating. Furthermore, PRs in this study reported enjoying and benefitting from interacting with fellow PRs. Therefore, forming a community within the raising group, similar to DeWitt’s [21] findings, may increase the program’s PR retention rate in the future.

Most PRs were excited to learn and practise their puppy-raising skills for their own enjoyment and future use if they got their own puppy. This is consistent with the ‘understanding’ function of the FAV [26]. However, due to the complex nature of raising puppies, PRs’ other motivational functions, namely ‘values’ and ‘enhancement’, relied on being taught the necessary skills to achieve their motivational reasons for being a PR (i.e., raising a successful puppy or enjoying living with the puppy).

In short, the deductive thematic analysis revealed that individuals became PRs to raise a puppy for someone in need, presuming they could also benefit from the experience of doing so. These altruistic and positive egoistic motivating factors attracted PRs and helped maintain their motivation throughout the program. However, it was apparent that without adequate support, PRs struggled to meet their motivational goals for being PRs, decreasing their motivation and, potentially, their willingness to raise subsequent puppies.

### 4.2. Limitations and Future Directions

A potential limitation of the current study might have been interviewer bias, resulting in unfair prejudices that may have altered the interview’s outcome or the interviewees’ responses [45]. Specifically, the author D.M. conducted all the interviews at both campuses. However, D.M. was based at the Bendigo campus and the training sessions were conducted within the research facility where he worked. As part of the field research design for this pilot cohort, author D.M. conducted informal observations and engaged with the trainers, PRs, and socialisers on a daily basis, during which there were no data collected for those informal activities. To address this potential interviewer bias, we created a semi-structured interview schedule as part of the ethics application, before D.M. met any of the participants. After the interviews, we conducted cross-checking with co-authors P.C.B., P.B., and T.J.H. during weekly research meetings. Therefore, interview bias was not a specific consideration during the data collection period. Future research may consider employing a third-party interviewer who has had no interaction with the PRs or involvement with the study, understanding that this approach may not benefit from the rapport and insight gained from engaging with the participants.

We did not ask PRs about their intention to return to raise a subsequent puppy after completion of their first program, as this was beyond the scope of the current study. Future research should aim to ask returning PRs about the specific reasons for why they return; this will help AD organisations further optimise their programs, thus increasing their retention rates. Similarly, we were unable to compare the suitability of the Bendigo and Bundoora campuses as AD puppy-raising facilities, but future research should aim to determine whether regional or metropolitan campuses are preferable for programs of this kind. Finally, comparing the experiences of staff and students was beyond the scope of the current study, but it is a direction for further research.

Using a phenomenological approach, the primary focus of this study was on the raisers’ experiences. Naturally, all volunteer raisers in the current study expected their puppy to successfully qualify as assistance dogs. Although puppy-training success is a common outcome measure in AD research, Mai et al. [20] referred to the puppy-raising program as a general system, arguing that puppy raisers’ experiences and competency were central to the overall success of the entire puppy-raising program. As outlined in Mai et al.’s [20] general puppy-raising model, examining the effect of the program on puppy qualification outcomes would require a separate study with measures of the puppy’s baseline characteristics, plus both raiser- and puppy-specific outcome measures, to arrive at any valid conclusions regarding the effectiveness of a PR program on the puppy’s qualification outcomes. Therefore, commenting on puppy outcomes would be beyond the scope of a phenomenological study (i.e., focusing on experiences).

As explained previously, the aim of this study was specifically focused on the experiences of puppy raisers and did not investigate the welfare of the puppies or their behavioural development. The general public typically believes that assistance dogs experience good welfare [2], but this is not guaranteed and empirical evidence is limited [46]. Therefore, future research should focus more attention on assistance dog welfare and development.

Whilst this study did achieve its intended goal of understanding how the AD program impacts PRs’ experiences, future research may build upon our findings by employing a quantitative survey with appropriate statistical analyses to examine what types of support are helpful for PRs, depending on demographics (e.g., age, occupation, household members), and how the wider community affects the PR experience. Ideally, this could show future organisations the kind of support that is most helpful for PRs, depending on their demographics, life circumstances, and competency levels. Although we found a range of social and psychological benefits from a campus-based puppy-raising program, they all came from the same program provider and the same higher education institution. Therefore, some of those benefits may be exclusive to puppy-raising programs at higher education settings or perhaps they are widely available in other private corporations. Future research is needed to evaluate the economic effectiveness as well as social and psychological outcomes of PR programs coming from those collaborations. By applying those quantitative measures to future AD programs in different settings (e.g., universities, private corporations) and regions, future research may be able to identify ideal candidates and locations for raising successful AD puppies.

## 5. Conclusions

The current study aimed to explore what factors impact the experiences of puppy raisers (PR) in assistance dog (AD) puppy-raising programs. The results showed that organisational support substantially impacted PRs’ raising abilities, while the practical support they received helped ease the difficulty of raising a puppy independently. The findings from both campus groups indicated that fewer challenges were associated with higher PR competency and elevated levels of enjoyment, with altruistic and egoistic motivational reasons attracting PRs to the program. Furthermore, as long as enough of the PRs’ motivational reasons were satisfied, they maintained motivation, enjoyed their experiences and were willing to adapt to and overcome puppy-raising challenges. Our findings add to the small body of literature pertaining to PRs, potentially helping to inform future AD programs and researchers on how to optimise PR experiences, including what methods could be applied to ensure volunteers enjoy their experience as PRs.

## Figures and Tables

**Table 1 animals-13-01482-t001:** Weeks Between the PRs Receiving Their Puppy and Attending Interviews.

Participant	Interview Number									
	1	2	3	4	5	6	7	8	9	10
**B1**	**Pre**	W1 *	**W5 ***	W16	**W23**	W31	-	-	-	W35
**B2**	**Pre**	W1 *	**W8**	W12	**W21 ***	W28	W32	-	-	W51
**B3**	**Pre**	W1 *	**W14**	**W20 ***	W27	W31	W40	-	-	-
**B4**	**Pre**	W1 *	**W4**	W8	**W22**	W26 *	W33	-	-	W37
**B5**	**Pre**	W2 *	**W4 ***	**W20**	W27	W34	W44	-	-	W59
**B6**	**Pre**	W1 *	**W4**	W8	**W23**	W32	W40 *	-	-	W56
**B7**	**Pre**	W1 *	**W4**	W9	W12	**W21**	W28	W35 *	-	-
**B8**	**Pre**	W1 *	**W4**	W8	W16 *	**W22**	W29	W39	W46	W50
**U1**	**Pre**	**W4**	**W19**	-	-	-	-	-	-	-
**U2**	**Pre**	**W4**	**W25**	-	-	-	-	-	-	-
**U3**	**Pre**	**W5**	**W24**	-	-	-	-	-	-	-
**U4**	**Pre**	**W5**	**W27**	-	-	-	-	-	-	-
**U5**	**Pre**	**W4**	**W30**	-	-	-	-	-	-	-
**U6**	**Pre**	**W7**	**W30**	-	-	-	-	-	-	-
**U7**	**Pre**	**W5**	**W31**	-	-	-	-	-	-	-
**U8**	**Pre**	**W7**	**W25**	-	-	-	-	-	-	-

Note. * interviews included in Mai, Howell [33]. B = PRs from the Bendigo group; U = PRs from the Bundoora group; Pre = initial interviews prior to PRs receiving their puppy; W = number of weeks since PRs received their puppy; boldface (e.g., **Pre** for all participants) = interviews selected for analysis; greyed (e.g., W1 of B1) = available data not selected for analysis; dash = data not available.

**Table 2 animals-13-01482-t002:** Main Themes, Descriptions of Sub-Themes, and Relevant Quotes from Puppy Raisers.

Main Theme	Sub-Theme	Explanation of Sub-Theme	Illustrative Quotes
Expectations as a PR	Expectation of challenges	Challenges that PRs expected to encounter when raising their puppies	“A few times where they’re a bit annoying, keep barking and all that stuff, but that’s the point of training them, so it makes them better dogs. It’s not always a good time, but most of the time it outweighs the bad.” (B4)
	Organisational expectations	The kinds of support PRs expected to be receiving from the organisation	“Basically, just to help the dog and just connect me to the dog to be able to raise it in the best way possible for when it does leave me. That’s what I would expect.” (U7)
	Household and work expectations	Support that PRs expected to be receiving from their household members and work colleagues	“My housemate is interested in helping out as well, so that’s going to be a help, not a hindrance.” (B2)
Organisational and Environmental Support and Suitability	Organisation informational support	The effectiveness of informational support that helps PRs live with and raise their puppy	“[An organisation trainer] was the one that has been giving me advice on how to manage [puppy’s] anxiety. She’s been very helpful because it was her suggestion to use the crate in the office more, it’s been really helpful.” (U5)
	Organisation training support	The effectiveness of training support that PRs received from the program	“[PRs puppy] went backwards quite significantly because [an organisation trainer] made me give her treats for things that I considered really bad behaviour by [puppy], but she kept telling me that she knew best. That was a bit disappointing.” (U4)
	Puppies influencing raisers	PRs are affected by puppies’ behavioural outcomes	“She is a good dog, it’s just it can be difficult and stressful too, because you’re like I mustn’t be very good with what I’m doing if I can’t get her to drop this shoe, something like that.” (B8)
	Home, work, and community suitability	PRs’ home and work environment helped and hindered PRs’ ability to effectively train their puppies	“Where I live is pretty good in terms of different locations. I can take her to places that I know will be quiet or I can take it to places that I know will be really busy…the campus and where I live has been pretty good for all that.” (U6)
Intensity of PRs Workload	Acceptance of difficulty	PRs learnt to accept some of the difficulties of being a PR	“We talked to the trainers about how to manage this. Some things just take time and there might not be a specific intervention that will immediately solve the problem. It’s like, well, I’m doing the right thing. I just have to keep doing it.” (U2)
	Conflicting demands	PRs led busy lives and struggled to dedicate sufficient time to training their puppies	“To be honest like it’s quite a big thing on top of a big load. I work four days a week, have a pretty big family, quite an intense job and so it’s a big project on top of that.” (U3)
	Need more support	PRs needed more support to benefit both themselves and their puppies	“I wouldn’t mind a socialiser or two. I wouldn’t mind a break every now and then. It’s a lot having a puppy 24/7.” (U5)
Practical Support	Socialisers	Socialisers helped and hindered the PRs and their puppies	“I think for the socialisers to be effective, not only do they need to be taking the dog out once a week, or whatever, they need to be going to the training. Without those things, they’re not really effective or helpful.” (U2)
	Peers	PRs interacting with each other benefitted the PRs and their puppies	“The interactions are really important between me and the other puppy raisers and [puppy] and other dogs… because it’s just another thing that they could be exposed to. The more we expose them, the better they will be.” (B8)
	Household and work	PRs household members and work environments help them look after and train their puppy	“By having my staff come to the training… they’re trained, and everybody’s familiar with what needs to be done so they can practice [the puppy’s] training.” (B5)
Motivations to Participate	Values	PRs were able express their altruistic and humanitarian values	“I would love to help train these puppies. Sorry, that’s so strange. I just would love to train the puppies. I would get such satisfaction out of training them for a good cause.” (B4)
	Enhancement	PRs were able to serve the aspect of their ego centred on personal growth and positive development.	“[An organisation trainer] says, ‘I’m the expert but I can tell you that you’re doing a good job’, you feel very good about yourself. You feel very good about the relationship because you’re not anxious and you’re not letting each other down.” (B1)
	Social	PRs were able to develop new relationships within and outside their raising group	“It’s like a real community when we meet up together on the Friday and we talk about any problems we’ve had or if someone’s dog has gone to the toilet in the shopping centre, we all kind of laugh about it and give them sympathy.” (B2)
	Understanding	PRs were able to enhance their puppy-raising knowledge	“It’s been really interesting learning all the training stuff as well, so all the different techniques.” (U6)
	Career	PRs were able to progress their career by being a raiser in the program	“Then I really hope the program works out for me if I get into it, because as I said it would be perfect for my career in the future.” (B8)

## Data Availability

The data are not publicly available due to privacy and ethical reasons.

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
