# Peer review of "Exploring the Experiences of Volunteer Assistance Dog Puppy Raisers from the Same Program at Two Australian University Campuses"

_animals, 2023, doi:10.3390/ani13091482_

Round 1

Reviewer 1 Report

The article is generally really well written and was a joy to read.

I don’t have any major concerns, just a few specific recommendations.

Simple summary:

·        Line 19 Australia not “Australian”

·        You say interviewed “repeatedly”, and later you say, “multiple times” and “several follow-up interviews” – could you just say the exact number to be clearer? Both in the abstract and later in the material section. Its not until line 219 that you say its between six to 10 interviews for Bendigo group, and 3 for the Bundoora group.

Introduction

·        You start with “Assistance animals” and then move to “Assistance dogs”. There is no linking to explain to someone that doesn’t know anything about this topic, so think more is needed. For example, explain that animals are increasingly being recruited to provide support to people (e.g. people with disabilities, older people), and provide a bit of background about their role, prior to talking about assistance dogs.

·        In terms of puppy raising programs – is it also worth mentioning that puppies can’t be left for more than three hours a day, which places additional demands on volunteers? Also can they go everywhere initially, as the training requires increased exposure over time? I think this would help explain a bit more about the “cost” of volunteering which you mention. As I don’t think its financial that you are meaning, its is more about the time factor.

·        Line 94 “PRs have over first-time PRs” – delete the last PRs.

·        Line 130 – not sure there is a need for a new paragraph as this is following on from line 129 and the Chur-Hanset et al study.

·        Was an aim not also the suitability of the university environment for running AD raising programs? This comes through in the results section.

Method

·        It isn’t clear why there was a difference in the number of interviews that the Bendigo group and Bundoora group attended. And also how it was determined the week when they attended interviews – was this pre-determined, or just due to availability of the participants.

Results

·        “organisational support in their second interviews than in their third interviews” – is there a reason for this, did the support increase from the organisation?

·        You mention “Misunderstanding AD laws” and the problems it caused, for one PR it seems particularly distressing. You mention in the introduction the dogs need to pass their training to be granted the access rights of ADs, which makes it sound like during training they don’t have access rights, not sure that’s the case, and seems like your research supports findings of another study: a survey of assistance animal users found differing levels of public awareness or acceptance of assistance animals by the general community; and pointed to the need for greater public awareness.  I don’t think in the discussion you mention this point, but I think its worthy of some discussion.

·        Is there a reason why Bundoora PRs reported intensity of workload more than Bendigo? Is there a better community, more support, being smaller? Maybe not a point to discuss here, but if you add the extra aim, could be a discussion point?

·        I am not sure I understand who the socialisers were – are they from the organisation? Again there are differences between Bundoora and Bendigo coming through that could be worth discussing in relation to the suitability of the university environment for running AD raising programs? Similarly in lines 355-356 you talk about PRs from the Bendigo campus having more social benefits and interacting more.

·        Line 341 “when the PRs were unavailable” – when are PRs not available, could you include i.e. reason 1, reason 2.

·        Not sure what you mean by “However, these statements focused on the positive aspects of the ego rather than acting as a protective process” – more explanation would be helpful.

Discussion

·        Lines 377 and 378 state some expected benefits, and this all seemed like new information, I can’t see where this is mentioned in the results. This section should be a discussion of the results, the so what part.

·        Line 396 should it be ‘reduce’ and ‘manage’ to keep the tense the same as line 396 (increase)?

·        If you add the aim to understand the suitability of the university environment for running AD raising programs then I think there would be additional discussion points you could comment on in this section. Seems like the university environment offered many benefits that maybe aren’t possible elsewhere, but there were also differences between Bundoora and Bendigo university environments?

Limitations

·        You raise a point about potential interviewer bias, but I’m not sure I understand what interaction the researcher had with PRs during the study outside of the interviews? Were they part of the training?

·        You earlier stated that you hoped the results would lead to more satisfactory volunteer experiences and entice them to raise subsequent puppies, so I was somewhat surprised that the interview didn’t cover this point. Maybe could rephrase and state that you were specifically interested in the experiences of being involved in a AD program, and therefore didn’t ask about their intention to return, rather than it being “beyond the scope”.

·        Was it the exploratory nature that limited the sample size? I think you are perhaps being a bit too harsh on yourself here. Firstly, it’s a new program that you are piloting, so makes sense to be small. Secondly, 16 participants is not that small a sample. I also don’t think larger samples and quantitative methods equals more empirical evidence. With larger samples you often end up talking in averages, losing some “outliers” that provide rich data.  Your qual study has highlighted some important key differences that you could talk about more e.g. differences across the two campuses; how some people had negative experiences. That’s not to say more research isn’t required, as it would be good to explore the additional factors you mention, but don’t think you need to be negative about your methods and the benefits it brings. There are pros and cons of all methods, but yours seemed fit for purpose.

·        “By applying this survey to…” but you used an interview?

Conclusion

·        Do you need to spell out PR and AD again, those acronyms have been used throughout the paper.

Author Response

The article is generally really well written and was a joy to read.

I don’t have any major concerns, just a few specific recommendations.

Simple summary:

  • Line 19 Australia not “Australian”

We’ve revised as suggested.

  • You say interviewed “repeatedly”, and later you say, “multiple times” and “several follow-up interviews” – could you just say the exact number to be clearer? Both in the abstract and later in the material section. Its not until line 219 that you say its between six to 10 interviews for Bendigo group, and 3 for the Bundoora group.

We’ve now added those details in the simple summary, as below (L20-21):

We interviewed 16 participants throughout the program (3 times for the Bundoora campus, and 7-10 times for the Bendigo campus) and analysed the data thematically to understand their experiences.

Introduction

  • You start with “Assistance animals” and then move to “Assistance dogs”. There is no linking to explain to someone that doesn’t know anything about this topic, so think more is needed. For example, explain that animals are increasingly being recruited to provide support to people (e.g. people with disabilities, older people), and provide a bit of background about their role, prior to talking about assistance dogs.

We’ve added a sentence to suggest that assistance animals refer to the assistance roles instead of the species that carry out the roles and the word “specifically” to suggest that assistance dogs refer to the species (i.e., dogs) that assume the assistance role. (L48-52)

The term ‘assistance animals’ refers broadly to the role and training requirements of the animals instead of their species (e.g., equine or canine)[3]. Following specialised training, they can accompany their handler into public spaces prohibited to most animals [3]. Assistance dog (AD) is an umbrella term for dogs specifically that are assistance animals and have been trained for one of several roles, including, but not limited to, guide, hearing, and alert dogs [3].

  • In terms of puppy raising programs – is it also worth mentioning that puppies can’t be left for more than three hours a day, which places additional demands on volunteers? Also can they go everywhere initially, as the training requires increased exposure over time? I think this would help explain a bit more about the “cost” of volunteering which you mention. As I don’t think its financial that you are meaning, its is more about the time factor.

We have incorporated this suggestion. Please refer to Lines 107-109.

The cost associated with the puppy raising program also includes the time and efforts of the puppy raisers especially when volunteer PRs are not allowed to leave the puppies unattended for more than 3 hours.

  • Line 94 “PRs have over first-time PRs” – delete the last PRs.

We have revised as suggested.

  • Line 130 – not sure there is a need for a new paragraph as this is following on from line 129 and the Chur-Hanset et al study.

We have merged the two paragraphs as suggested.

  • Was an aim not also the suitability of the university environment for running AD raising programs? This comes through in the results section.

We thank the reviewer for this suggestion. We have added this as a secondary aim in the section ‘The Current Study’. (L164-166)

Since the programs were implemented at two university campuses, a secondary aim of this study was to understand the suitability of campus environments for running AD raising programs.

Method

  • It isn’t clear why there was a difference in the number of interviews that the Bendigo group and Bundoora group attended. And also how it was determined the week when they attended interviews – was this pre-determined, or just due to availability of the participants.

There was not much known about the feasibility of operating a campus-based puppy raising program. Therefore, we had to pilot it at a smaller campus, i.e., Bendigo, during which time, it was necessary to extensively conduct more interviews to capture any potential issues with this new model. The interviews with the Bundoora group followed a more common longitudinal qualitative research design, i.e., 3 times. We have explained this in the manuscript (L207-216).

Because implementing a puppy raising program as part of the university’s approved activity was not widely known, the program was piloted at the smaller campus, i.e., Bendigo, before being rolled out to the Bundoora campus. The two programs follow the same puppy raising protocols from the AD provider. However, the research designs varied slightly to reflect the piloting and normal implementation nature of the overall project; there were also different research ethics approvals for the two cohorts. Specifically, during the piloting phase, we conducted more frequent interviews (7-10 times with the Bendigo PRs), while the main program at the Bundoora campus followed a more common research design with interviews conducted at 3 time points.

Results

  • “organisational support in their second interviews than in their third interviews” – is there a reason for this, did the support increase from the organisation?

There are two possible explanations for the perceived inadequate organisational support. One could be that the demands for managing the puppy’s learning reduced as they developed, or that the raisers would gain more realistic expectations of what the organisation may be able to provide within their resources. We have explained this in the manuscript (L318-322)

PRs mentioned more problems stemming from a lack of organisational support in their second interviews than in their third interviews. The expressed unmet expectations of organisational support might be due to higher demands for managing the puppies’ behaviours in the earlier stage (second interview) or it could be that the PRs had higher expectations of the organisational support which they might have re-calibrated as they gained more experience in this role (third interview).

  • You mention “Misunderstanding AD laws” and the problems it caused, for one PR it seems particularly distressing. You mention in the introduction the dogs need to pass their training to be granted the access rights of ADs, which makes it sound like during training they don’t have access rights, not sure that’s the case, and seems like your research supports findings of another study: a survey of assistance animal users found differing levels of public awareness or acceptance of assistance animals by the general community; and pointed to the need for greater public awareness. I don’t think in the discussion you mention this point, but I think its worthy of some discussion.

We thank the reviewer for this comment. We have added a short discussion on this point (L466-476)

The benefits of having a shared understanding with others were more apparent during the current campus-based puppy raising programs, especially when it came to the legal understanding of public access rights of the assistance animals and those in training. The PRs in the current study reported positive experiences when interacting with colleagues and the university community during socialisation and training of their puppies, even though they reported being denied access when conducting those activities in public places. Although public access for ADs in training varies by jurisdiction [42], in many parts of Australia, including the state of Victoria, public access for AD puppies is permitted [43]. Nonetheless, PRs commonly reported that general community misunderstanding about access rights of AD puppies in training was deterring and discouraging [44].

  • Is there a reason why Bundoora PRs reported intensity of workload more than Bendigo? Is there a better community, more support, being smaller? Maybe not a point to discuss here, but if you add the extra aim, could be a discussion point?

It was an interesting point that we also initially considered. However, with the lack of data about why, we could not incorporate this into our findings. Our understanding is that the program was almost identical, if not more supported at the Bundoora campus, as the program provider gained some experience from running the program at the Bendigo campus. We were tempted to theorise that the Bundoora PRs lived in the major metropolitan area and therefore the just had a busier lifestyle. The amount of time they anticipated as required for the program may have been less than the Bendigo PRs. However, these were all our guesses with no specific data to back them up. Due to the lack of data regarding this point, we decided not to include it in the original submission. We did add a short statement to L348-349:

The reasons for this discrepancy are not clear from the available data.

  • I am not sure I understand who the socialisers were – are they from the organisation? Again, there are differences between Bundoora and Bendigo coming through that could be worth discussing in relation to the suitability of the university environment for running AD raising programs? Similarly in lines 355-356 you talk about PRs from the Bendigo campus having more social benefits and interacting more.

We have described those volunteer socialisers and their role in the manuscript (L356-362)

In the current study, socialisers were recruited from the university community (i.e., staff and students), who would like to be involved but did not have a suitable living environment to raise the puppy or they simply could not provide long-term care for the puppy. They attended training with the PRs and were coordinated by staff from the AD provider, to take over the socialisation activities for the puppies around the campus for a couple of hours per week while the PRs were working or studying on campus.

  • Line 341 “when the PRs were unavailable” – when are PRs not available, could you include i.e. reason 1, reason 2.

We have clarified that statement, L369-372.

Instead, most PRs reported that individuals in their homes and workplaces often actively assisted in training, taking the puppy for walks, and even caring for them when the PRs took some time off or visited places where they did not want to bring the puppies.

  • Not sure what you mean by “However, these statements focused on the positive aspects of the ego rather than acting as a protective process” – more explanation would be helpful.

We have revised this sentence with more clarification (L380-383)

However, these statements refer more to their altruistic intent and a sense of personal growth rather than acting as a protective process to reduce negative feelings resulting from guilt of being more fortunate than others.

Discussion

  • Lines 377 and 378 state some expected benefits, and this all seemed like new information, I can’t see where this is mentioned in the results. This section should be a discussion of the results, the so what part.

We have taken that statement out of the discussion.

  • Line 396 should it be ‘reduce’ and ‘manage’ to keep the tense the same as line 396 (increase)?

We have revised writing of that sentence (L429-431)

Our findings suggest that PRs wanted organisational support to increase their competency, which would reduce the difficulty of training and enable them to better manage their puppy’s behaviours.

  • If you add the aim to understand the suitability of the university environment for running AD raising programs then I think there would be additional discussion points you could comment on in this section. Seems like the university environment offered many benefits that maybe aren’t possible elsewhere, but there were also differences between Bundoora and Bendigo university environments?

We have added a paragraph with a statement to suggest the suitability of the university environment more explicitly. (L476-483)

Overall, with the convenience of conducting puppy training and socialisation at workplace (for staff) and their learning environment (for students), along with free access to short-term carers (i.e., volunteer puppy sitters, or socialisers), university campuses appear to be a highly suitable environment in which AD provide should consider operating their puppy raising programs. However, the puppy raising program was fully supported by the University, which created a welcoming environment that may not be possible without full endorsement by university administrators.

As mentioned above, because we do not have data and a relevant a priori aim, we could not make any explicit comparison of suitability between the two campuses in this paper. However, it is an interesting point that we are also eager to investigate in our future research. Thus, we added the following to L537-540:

Similarly, we were unable to compare the suitability of the Bendigo and Bundoora campuses as AD puppy raising facilities, but future research should aim to determine whether regional or metropolitan campuses are preferable for programs of this kind.

Limitations

  • You raise a point about potential interviewer bias, but I’m not sure I understand what interaction the researcher had with PRs during the study outside of the interviews? Were they part of the training?

We have added further clarification for the potential interviewer bias which we attempted to minimise. (L519-531)

Specifically, author D.M conducted all the interviews at both campuses. However, D.M was based at the Bendigo campus and the training sessions were conducted within the research facility where he worked. As part of the field research design for this pilot cohort, author D.M. conducted informal observations and engaged with the trainers, PRs and socialisers on a daily basis, during which there was no data collected for those informal activities. To address this potential interviewer bias, we created a semi-structured interview schedule as part of the ethics application, before D.M. met any of the participants. After the interviews, we conducted cross-checking with co-authors P.C.B., P.B., and T.J.H. during weekly research meetings. Therefore, interview bias was not a specific consideration during the data collection period. Future research may consider employing a third-party interviewer who has had no interaction with the PRs or involvement with the study, understanding that this approach may not benefit from the rapport and insight from engaging with the participants.

  • You earlier stated that you hoped the results would lead to more satisfactory volunteer experiences and entice them to raise subsequent puppies, so I was somewhat surprised that the interview didn’t cover this point. Maybe could rephrase and state that you were specifically interested in the experiences of being involved in a AD program, and therefore didn’t ask about their intention to return, rather than it being “beyond the scope”.

The statement that the reviewer referred to was in the section The Current Study. We believe the current phrasing was ambiguous which created some confusion. We have now rephrased it so it appropriately refers to the long-term outcomes of the current findings, instead of what we expected from the PRs in this study. (L167-169)

The findings of this study will hopefully help future PR programs to improve their PRs’ overall program outcomes, with the long-term benefits for the AD industry including leading to more satisfactory volunteer experiences and enticing them to raise subsequent puppies.

  • Was it the exploratory nature that limited the sample size? I think you are perhaps being a bit too harsh on yourself here. Firstly, it’s a new program that you are piloting, so makes sense to be small. Secondly, 16 participants is not that small a sample. I also don’t think larger samples and quantitative methods equals more empirical evidence. With larger samples you often end up talking in averages, losing some “outliers” that provide rich data. Your qual study has highlighted some important key differences that you could talk about more e.g., differences across the two campuses; how some people had negative experiences. That’s not to say more research isn’t required, as it would be good to explore the additional factors you mention, but don’t think you need to be negative about your methods and the benefits it brings. There are pros and cons of all methods, but yours seemed fit for purpose.

We have removed that statement to avoid the impression that we over criticised our current study design. The revision now refers to what future research may build upon our findings. (L542-546)

Whilst this study did achieve its intended goal of understanding how the AD program impacts PRs’ experiences, future research may build upon our findings by employing a quantitative survey with appropriate statistical analyses to examine what types of support are helpful for PRs depending on demographics (e.g., age, occupation, household members), and how the wider community affects the PR experience. Ideally, this could show future organisations the kind of support that is most helpful for PRs, depending on their demographics, life circumstances, and competency levels.

  • “By applying this survey to…” but you used an interview?

The current statement refers to future research instead of the current one. We have revised it to be specific. (L556-559)

By applying those quantitative measures to future AD programs in different settings (e.g., universities, private corporations) and regions, future research may be able to identify ideal candidates and locations for raising successful AD puppies.

 Conclusion

  • Do you need to spell out PR and AD again, those acronyms have been used throughout the paper.

We believe it was acceptable to spell out the acronyms per writing convention for conclusion sections.

Reviewer 2 Report

Thank you for the opportunity to review this paper. I found it scientifically sound and compelling. I saw no need for revisions. 

Author Response

I would like more explanation about the intervals of interviews between the two campuses. They do not seem to match and I am interested in what influenced those discrepancies.

There was not much known about the feasibility of operating a campus-based puppy raising program. Therefore, we had to pilot it at a smaller campus, i.e., Bendigo, during which time, it was necessary to conduct extensively more interviews to capture any potential issues with this new model. The interviews with the Bundoora group followed a more common longitudinal qualitative research design, i.e., 3 times. We have explained this in the manuscript (L207-216).

Because implementing a puppy raising program as part of the university’s approved activity was not widely known, the program was piloted at the smaller campus, i.e., Bendigo, before being rolled out to the Bundoora campus. The two programs follow the same puppy raising protocols from the AD provider. However, the research designs varied slightly to reflect the piloting and normal implementation nature of the overall project; there were also different research ethics approvals for the two cohorts. Specifically, during the piloting phase, we conducted more frequent interviews (7-10 times with the Bendigo PRs), while the program at the Bundoora campus followed a more common research design with interviews conducted at 3 time points. 

On page 6 (Data Analysis) they mention that they used discussion and "investigator triangulation" to reduce the risk of researcher bias although they also point out that there could have been interviewer bias that affected the data collection. I would like a more robust explanation of the difference between interviewer bias and researcher bias in this project. It appears the interviewer was an author (i.e., researcher). This encourages me to question the development of the interview schedule - can they include information in methods about the development of the interview schedule.

We have added further clarification for the potential interviewer bias which we attempted to minimise. (L519-531)

Specifically, author D.M conducted all the interviews at both campuses. However, D.M was based at the Bendigo campus and the training sessions were conducted within the research facility where he worked. As part of the field research design for this pilot cohort, author D.M. conducted informal observations and engaged with the trainers, PRs and socialisers on a daily basis, during which there was no data collected for those informal activities. To address this potential interviewer bias, we created a semi-structured interview schedule as part of the ethics application, before D.M. met any of the participants. After the interviews, we conducted cross-checking with co-authors P.C.B., P.B., and T.J.H. during weekly research meetings. Therefore, interview bias was not a specific consideration during the data collection period. Future research may consider employing a third-party interviewer who has had no interaction with the PRs or involvement with the study, understanding that this approach may not benefit from the rapport and insight from engaging with the participants.

The authors sort the data by campus for their analysis. This is reasonable, although they specifically mentioned that 8 PRs were university staff and 8 PRs were grad/post-grad students. I am curious why they mention this specifically and then do not follow up with the similarities or differences between the groups based on their staff or student roles.

We appreciate the reviewer’s curiosity. We agree that it would be interesting to see if there are any differences or similarities based on their roles, However, the details on the PRs’ role were primarily for reporting demographic purposes and it was not the intent of this study to investigate any differences. While beyond the scope of the current study, it is a potential avenue for future research. L540-541 now say:

Finally, comparing the experiences of staff and students was beyond the scope of the current study, but it is a direction for further research.

Reviewer 3 Report

This study aimed to explore the experiences of assistance dog puppy raisers, enabling better future retention of these volunteers by assistance dog organisations. Qualitative data were collected from 16 University based Puppy Raisers. Data were analysed thematically, and authors considered their findings within the Functional Approach to Volunteerism framework. Benefits identified mirrored those in existing research regarding the practical and organsiational benefits Puppy Raisers receive. The paper also suggested four functional motivations for the volunteers. This included altruism, egoism, interacting with others, and learning new skills.

The introduction is well written and expands on the existing literature on the topic. However, there are several limitations to the study including a small, self selected sample, potential interviewer bias, and single puppy raising approach (ie. University based scheme). There is also a lack of measures on outcome/success of the puppy raisers (ie did the puppies become qualified as an assistance dogs). Moreover, inclusion of measures of the puppy’s experience (e.g. welfare/behavioural development etc) may have enhanced the paper. While the study will be of interest to assistance dog organisations, generalisation and application of results is limited. However, these limitations are clearly stated in the paper, and the study is of value as an exploratory qualitative pilot, highlighting areas for future research in an important area.

Author Response

We thank the reviewer for commending our work.